# Integrative Insights into Philadelphia-like B-Cell Acute Lymphoblastic Leukemia: A Genetic and Molecular Landscape

**DOI:** 10.3390/diagnostics15030385

**Published:** 2025-02-06

**Authors:** Stacey Chuang, Alexandra Chu, Rodrigo Hurtado, Carlos A. Tirado

**Affiliations:** 1The International Circle of Genetic Studies Project, Stony Brook Chapter, Stony Brook, NY 11794, USA; stacey.chuang@stonybrook.edu (S.C.); alexandra.chu@stonybrook.edu (A.C.); joserhv12@gmail.com (R.H.); 2Department of Pathology, Renaissance School of Medicine, Stony Brook University, Stony Brook, NY 11794, USA

**Keywords:** B-ALL, cytogenetics, Ph-like B-ALL

## Abstract

Philadelphia-like chromosome acute lymphoblastic leukemia (Ph-like ALL) is a new subtype of B-ALL that was discovered in 2009 and recognized in the 2016 revision of the World Health Organization criteria under the classification of myeloid neoplasms and acute leukemia. This new subtype has an extremely poor prognosis compared to that for other subtypes of ALL, with a 41% five-year overall survival (OS) rate. Ph-like ALL is chemoresistant, with a high minimum residual disease (MRD) level after induction therapy, and it is associated with a high relapse rate. Clinical trials are currently being conducted to study the effectiveness of specific tyrosine kinase inhibitors against different genetic alterations in Ph-like ALL patients and the effect of allogeneic hematopoietic cell transplants (allo-HCT) on treatments. This review summarizes the current findings on Ph-like ALL, focusing on its molecular landscape and clinical implications.

## 1. Introduction

Ph-like B-ALL (*BCR::ABL1*–like ALL) is a B-cell precursor ALL characterized by different genetic rearrangements and mutations [1]. Although Ph-like ALL is known as *BCR::ABL1*-like ALL, Ph-like refers to an ALL with molecular characteristics similar to those of Philadelphia-positive (Ph+) ALL, whereas BCR::ABL1-like ALL signifies that there is no BCR::ABL1 fusion. Most cases are concordant [2]. In this review, the term Ph-like will be used. Ph-like has a leukemic gene cell expression similar to that of Ph+ ALL but lacks the *BCR::ABL1* fusion from t(9;22)(q34;q11.2), an expression associated with a poor prognosis in Ph+. Ph-like ALL is a high-risk subtype of B-ALL, and it is not uncommon, with a higher prevalence in younger adults and Hispanics [1]. This descent is explained by a germline Ph-like ALL risk variant, *GATA3*. Ph-like is associated with higher MRD levels at the end of induction therapy and a poor outcome [1]. In Table 1, the prevalence of Ph-like B-ALL is indicated for each age group, with younger adults having the highest prevalence.

B-ALL typically has a good prognosis, especially for children, and treatments are improving for adults, with more effective treatments using pediatric-inspired regimens. Event-free survival (EFS) and OS have been reported to be 66% and 79%, respectively, for adults and over 90% for children [3]. However, Ph-like ALL has been observed to have a high disease relapse rate and poor overall survival compared to all the subtypes of ALL [4]. Young adults with the Ph-like form have the worst prognosis in comparison to children and adolescents [5].

## 2. Molecular Characterization

Ph-like ALL is a heterogeneous disease, mainly characterized by kinase-activating alterations. These alterations can be split into four general subgroups, with the most common alteration being JAK-STAT, which comprises rearrangements in *CRLF2*, *JAK2*, or *EPOR*. Other alterations are ABL-class fusions (*ABL1*, *ABL2*, *CSF1R*, *PDGFRA*, and *PDGFRB*), Ras pathway mutations (*NRAS*, *KRAS*, *PTPN11*, and *NF1*), and uncommon or unknown fusions [1]. *CRLF2* rearrangements involving *JAK2* wild are the most prevalent in all age groups (Figure 1). These diverse genetic alterations, with multiple kinase and cytokine receptors, make tyrosine kinase inhibitors (TKIs) a viable treatment option. The International Consensus Classification (ICC) divides cases of Ph-like ALL into subgroups: *ABL*-1-class-rearranged, JAK-STAT-activated, and NOS [6].

## 3. *CRLF2* Gene Rearrangements

The cytokine receptor 2 (*CRLF2*) is located on chromosomes Xp22.33 and Yp11.3 and has four transcript versions—two with eight exons, one with seven, and one with nine [7]. The *CRLF2* gene, located on chromosomes X and Y, generates nine exons after transcription (Figure 2). The cytokine receptor 2 (*CRLF2*) gene encodes a transmembrane protein that pairs with interleukin-7-receptor-α (*IL7R-α*), which heterodimerizes to form a receptor for thymic stromal lymphopoietin (*TSLP*) [8]. Upon the binding of the encoded protein to TSLP, three pathways are initiated, namely, JAK2, STAT3, and STAT5, to activate *CRLF2* expression (Figure 3) [9]. The pathways control cell proliferation and development in the hematopoietic system, processes that are associated with increased signaling activation when the *CRLF2* gene is mutated or rearranged [10]. *CRLF2* rearrangements and mutations are often found in association with poor prognoses and high rates of MRD after induction therapy [2]. In a study conducted by Harvey et al. in 2010, a cohort of patients was studied over the span of four years, and the relapse-free survival rates for *CRLF2* overexpression and *CRLF2* non-overexpression were determined to be 35.3% and 71.3%, respectively [11]. 

*CRLF2* alterations account for approximately 50% of Ph-like ALL cases and are generated by three different molecular mechanisms (Figure 4). One rearrangement is a small interstitial deletion of the pseudo-autosomal region (*ASMTL*, *SLC25A6*, *IL3RA*, and *CSF2RA*), which leads to *P2RY8::CRLF2* fusion, and this is more commonly found in young children [2,13]. This rearrangement leads to the joining of *CRLF2* to the first non-coding exon of *P2RY8*, which places the expression and activation of the pathways under control of the *P2RY8* promoter [2]. This fusion can be identified through reverse transcription polymerase chain reaction or genomic PCR that can show the chromosomal breakpoints [11]. *CSF2RA::CRLF2* has also been reported to fuse due to the deletion of *PAR1*, resulting in characteristics similar to those of *P2RY8::CRLF2* [2].

The second rearrangement is a V(D)J break at the IGH locus and a CpG break at *CRLF2*, creating *IGH::CRLF2*, t(X;14)(p22;q32), or t(Y;14)(p11;q32) [14]. It is associated with a poorer prognosis than that for *P2RY8::CRLF2*, with a higher *CRLF2* expression rate, and it is also associated with a higher level of minimum residual disease (MRD > 10^−3^) [14,15]. This fusion places *CRLF2* activation under the control of the *IGH* enhancer, leading to homodimerization at the cellular membrane, which over-activates *CRLF2* expression and its associated pathways (JAK2/STAT5, PI3k/MTOR, and SRC), which leads to lymphoblast proliferation and survival [14]. This rearrangement is more commonly seen in adolescents or young adults [16]. This fusion can be identified via FISH [2].

**Figure 4 diagnostics-15-00385-f004:**
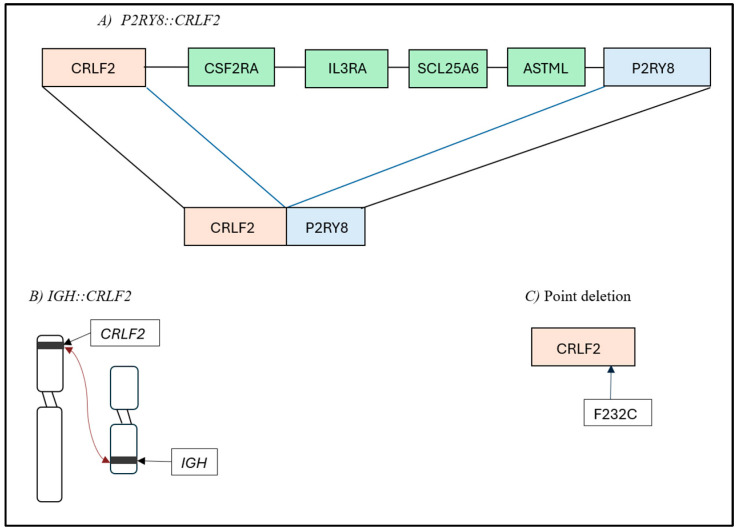
(**A**) *P2RY8::CRLF2*, with interstitial deletion of the *PAR1* region shown in green. (**B**) *IGH::CRLF2*, with a cryptic translocation of *CRLF2* and *IGH*. Diagram shows t(X;14)(p22;q32), but translocation can also be t(Y;14)(p11;q32). (**C**) Point deletion of *CRLF2*; the most common one is depicted. Diagrams A and C were adapted from [2], and B was adapted from [17].

The third, rarer rearrangement is a *CRLF2* point mutation (single-nucleotide variants), which frequently occurs on F232C [10]. This point mutation promotes homodimerization, which stimulates cytokine growth [2].

About 50% of *CRLF2* rearrangements have Janus kinase 1 or 2 (*JAK1*/*JAK2)* point mutations. *CRLF2* overexpression is more commonly associated with *JAK2* point mutations [16]. The most common *JAK2* point mutation is R683S (G). *JAK2* plays key roles in the hematopoietic and immune responses. *JAK2* R683S (G) mutations indicate B-all; however, the mechanism behind how the mutations lead to B-ALL is still unclear. Mutations in R683S (G) increase *JAK2* activity and impair the structure of the *JAK2* domain, leaving it to become partially unfolded. Constitutive activation has been suspected to occur because of the partially unfolded state [18].

## 4. *ABL* Class Rearrangements

This subgroup of Ph-like ALL makes up approximately 15% of all Ph-like ALL cases, with a higher prevalence in children than adults [19]. The presence of these translocations indicates Ph-like ALL, and they are usually associated with *IKZF1* mutations and deletions. The loss of *IKZF1* function leads to uncontrollable lymphoid progenitor self-renewal due to the loss of negative regulation [20]. *IKZF1*, located on 7p12.2, is composed of eight exons that encode the IKAROS protein, a transcription factor with zinc-finger-type DNA-binding domains [20]. IKAROS has six zinc finger domains. Four of them, located at the N-terminal end and encoded by exons 4–6, allow its binding to specific DNA sequences in the genome. On the other hand, exon 8 encodes two zinc fingers that facilitate the formation of homodimers or heterodimers (Figure 5) [21]. *IKZF1* has been shown to regulate gene expression through three main mechanisms: modifying chromatin structure by interacting with remodeling complexes; enhancing and interacting with the RNA polymerase II transcription initiation complex; and promoting changes in chromosome conformation [22]. The ability of IKAROS in its wild-type form to form homo- or heterodimers with other members of its family is essential for the localization and regulation of pericentromeric heterochromatin (PCH), which plays key roles in genome stability, transposon suppression, and proper chromosome segregation. However, mutations in *IKZF1* affecting the DNA-binding domains (ZF1-6) alter its dimerization capacity, reduce its affinity for DNA, and cause aberrant localization in the PCH, compromising its regulatory biological functions [23]. Alterations in IKAROS have been identified to be present in up to 68% of Ph-like ALL cases. Furthermore, clinical studies have shown that Ph-like ALL patients with *IKZF1* deletions have a lower five-year event-free survival than Ph-like ALL patients without *IKZF1* deletion [20].

*ABL* gene fusions consist of *ABL1* and *ABL2*, which are tyrosine kinases, and *CSF1R*, *PDGFRA*, and *PDGFRB*, which are cytokine receptor tyrosine kinases. *ABL* class genes are responsible for the pathophysiology of Ph-like ALL. Each *ABL* class gene has its own fusion partners, as shown in Table 2. All fusions follow the same process, where the 5′ fusion partner leads to the constitutive activation of the *ABL* 3′ gene [2].

## 5. JAK-STAT Rearrangements

Two common JAK-STAT alterations are *JAK2* and *EPOR* rearrangements, which make up 7–15% of all Ph-like ALL cases. They are associated with the worst prognosis among all subgroups [19]. *JAK2* rearrangements can occur with any of the twenty-eight fusion partners [2,24]. While the amino terminus is determined by the partner gene it is fused with, the *JAK2* 3′ tyrosine kinase domain is preserved and always active (Figure 6) [2,25]. Of the fusion partner genes, the most common are *PAX5*, *BCR*, *ETV6*, *SSBP2*, and *ATF7IP* [26]. Regardless of whether these fusions include the upstream pseudokinase domain (encoded by exons 13 to 18, as shown in Figure 7), they have been shown to activate STAT5 in vitro [2].

Several of these fusion partners were identified recently, including *GOLGA4*, *NPHP3*, and *STRBP*, which were discovered in 2021 [26,28]. The latest-identified fusion partner is *ZEB2*. In 2023, researchers discovered the *ZEB2-JAK2* fusion gene in a 42-year-old male patient from China using transcriptome sequencing [29]. The patient also exhibited a *PAX5* frameshift mutation and was diagnosed with refractory relapsed Ph-like B-ALL with *JAK2* rearrangement [29]. To date, JAK2 is known to have twenty-eight fusion partners, while EPOR has four. Both can be treated with ruxolitinib.

Other JAK-STAT alterations include changes in *IL7R*, *IL2RB*, *JAK1*, *JAK3*, *SH2B3*, *FLT3*, and *TYK2* (Table 3) [24,30,31], and these mutations are commonly multi-subclonal, meaning they are secondary driver events in Ph-like ALL [4]. While these mutations are common and may help drive the cancer, they typically emerge later, as the disease progresses, and are not the initial cause of Ph-like ALL. As such, they are considered secondary driver events—which are more often associated with multi-subclonality—rather than primary ones [4]. Constituting approximately 13% of Ph-like ALL cases, these alterations involve multiple genes that show mutations and DNA copy-number changes [4,5]. Mutations in *IL7R*, *FLT3*, and *IL2RB* permanently activate cytokine receptors as well as mutations in *JAK1* and *JAK3*, and the deletion of *SH2B3* leads to a loss of regulatory function and overactivation of the JAK-STAT pathway [5].

*EPOR*, located on chromosome 19p13.2, is normally responsible for the formation of the erythropoietin receptor. Erythropoietin is a hormone that regulates the production of erythrocytes in the bone marrow [32]. *EPOR* has four fusion partners: *IGH*, κ (*IGK*), leukocyte-associated immunoglobulin-like receptor 1 (*LAIR1)*, and thyroid adenoma-associated gene (*THADA)*. Though rearrangements involving *LAIR1* and *THADA* are less common than those involving *IGH* and *IGK*, translocations involving any of these four genes lead to abnormal expression of a truncated *EPOR* gene and EPOR protein [4]. Because these shortened *EPOR* proteins are missing negative regulatory domains, their stabilized expression on the surfaces of B cells activates JAK-STAT signaling and drives leukemogenesis [2,4].

## 6. Diagnostic Methods for Ph-like ALL

The diverse genomic landscape of Ph-like ALL has posed a significant diagnostic challenge to healthcare professionals. Patients with Ph-like ALL are classified according to CRLF2 expression, with those with high expression of this gene being evaluated by *CRLF2* rearrangement testing using fluorescence in situ hybridization (FISH) and *JAK1*/*JAK2* mutations via Sanger sequencing. On the other hand, patients with low CRLF2 expression are tested for other kinase alterations using reverse transcription polymerase chain reaction (RT-PCR), followed by transcriptome sequencing if PCR results are negative [1].

Chromosomal rearrangements in Ph-type acute lymphoblastic leukemia are often undetectable via conventional cytogenetics because they are cytogenetically cryptic. In response to this limitation, many laboratories around the world have implemented FISH analysis to identify common genetic translocations, such as those involving the *ABL1*, *ABL2*, *CRLF2*, *JAK2*, *EPOR*, *PDGFRB*, and *CSF1R* genes [32].

RT-PCR is an effective method for confirming genetic alterations detected in gene expression profiles through RNA sequencing. A quantitative model based on this technique that helps predict Ph-like cases has been developed, using a genetic signature that includes 10 overexpressed genes: *SOCS2*, *IFITM1*, *CD99*, *TP53INP1*, *IFITM2*, *JCHAIN*, *NUDT4*, *ADGRE5*, *SEMA6A*, and *CRLF2* [32].

Furthermore, increased staining of the thymic superficial stromal lymphopoietin receptor ((TSLPR) encoded by *CRLF2*) in ALL blasts, readily detectable through flow cytometry, has been shown to be a good indicator of *IGH::CRLF2* and *P2RY8::CRLF2* rearrangements as well as *CRLF2* F232 point mutations in Ph-like ALL cells. Clinical TSLPR immunophenotyping, now included in standard diagnostic flow cytometry panels, is highly efficient and allows the identification of patients with *CRLF2*-R B-ALL within the first 24 h after sample collection [31].

## 7. Treatment

Blinatumomab is an antibody drug that has been shown to be a considerably effective approach for patients with relapsed or refractory (r/r) Ph-like B-ALL or early persistent MRD. Blinatumomab is a CD3/CD19 antibody with a 75% complete response in r/r patients with *CRLF2* rearrangements and a 57% response in r/r patients with non-*CRLF2* rearrangements; however, more tests are needed to examine the benefits of using blinatumomab due to the limited sample sizes in previous studies [33].

Patients with JAK-STAT signaling, including cohorts with *CRLF2* rearrangements, *EPOR*, and *JAK2* mutations, can be subjected to treatments that use JAK inhibitors, such as Ruxolitinib [19]. Ruxolitinib inhibits the ATP-catalytic site on *JAK1* and *2*, which disrupts signaling pathways, leading to a decrease in cytokine levels [34]. In patients with *CRLF2* rearrangements, high dosages may be needed—a minimum of 50 mg administered twice daily for a clinical effect [19]. Additionally, Ruxolitinib can be used in cases with *SH2B3* deletions [19]. Some recently identified *JAK2* fusion partners are still awaiting the outcomes of current clinical trials evaluating the effectiveness of Ruxolitinib, but available evidence shows that JAK inhibitors should be used in treatment [26].

Patients with *ABL* class fusions can be treated with Dasatinib [1,32]. Dasatinib is a second-generation *ABL* kinase inhibitor that binds to both the active and inactive conformations of the *ABL* kinase domain, and it can cause remission and eliminate MRD in patients [2,35,36]. A 2023 study by Tan et al. assessed the efficacy of Dasatinib in three patients with *CNTRL::ABL1*, *LSM14A::ABL1*, and *FOXP1::ABL1* fusion genes. All three showed rapid, significant remission without any major adverse events and became the first patients with their respective fusions to be successfully treated with Dasatinib [37].

Allogeneic hematopoietic cell transplantation (allo-HCT), a procedure that replaces a patient’s destroyed blood stem cells with a donor’s healthy ones, is associated with a high mortality rate, so it is essential to select the appropriate patient [19]. It has resulted in a 50% three-year overall survival for adults with Ph-like ALL, but there is no significant difference in survival outcomes between Ph-like and non-Ph-like cohorts. Patients with MRD negativity show significantly better results, so efforts must be made to achieve this status before allo-HCT is carried out [19]. Relapse, especially in patients with *CRLF2* overexpression, is the main cause of unsuccessful allo-HCT procedures. Allo-HCT should be considered on an individual basis using criteria such as late MRD persistence, inadequate therapy, age, ability to undergo a transplant procedure with minimal risks, and a high relapse risk rate [19]. The effectiveness of allo-HCT is still being tested, and further research is needed to provide a more reliable set of criteria.

## 8. Discussion

Ph-like ALL is challenging to diagnose due to its heterogeneity, and it remains one of the highest-risk subtypes of B-ALL. Most molecular characteristics can be detected via FISH and PCR [15]. The most common, making up 42–60% of Ph-like cases, are rearrangements in *CRLF2* [38]. *CRLF2* rearrangements and point mutations promote STAT5 phosphorylation and activate JAK/STAT pathways, leading to ALL blast survival [12,39]. About 50–60% of patients with r-*CRLF2* also have genetic abnormalities of the JAK or RAS pathways, including *KRAS*, *NRAS*, *PTPN11*, and *NF1*, and these abnormalities may include the deletion of one allele of *IKZF1* [38]. These rearrangements are associated with a poorer prognosis, and the most common treatment is Ruxolitinib, a *JAK2* inhibitor [4].

The second most common is *ABL* class gene fusions, which are typically associated with *IKZF1* mutations or deletions. *ABL* fusions result in the formation of chimeric proteins, which lead to tyrosine kinase activities in genes where *ABL* is not abundantly expressed. This leads to the overactivation of pathways creating immature lymphoid cells [40].

*JAK2* and *EPOR* rearrangements, though not as common as the previously mentioned mutations in Ph-like ALL, are correlated with the poorest prognosis among all the subgroups, with a 23.5% 5-year EFS [41]. *JAK2* rearrangements can form fusion genes and proteins; the amino terminus is determined by the fusion partner (of which there are twenty-eight), while the *JAK2* kinase domain remains intact [2,24]. While germline *EPOR* mutations can result in autosomal-dominant benign erythrocytosis, truncated EPOR proteins activate JAK-STAT signaling [2]. Other JAK/STAT mutations do not show kinase or cytokine receptor gene rearrangements, but they often have chromosomal rearrangements that create fusion oncoproteins involving transcription factor genes and/or epigenetic regulators [4].

To conclude, Ph-like ALL is seen in approximately 20–30% of adolescents and adults with B-ALL, and it is associated with a five-year overall survival of 41% [42]. The purpose of this article is to provide an update on the current treatment progress and the different molecular characterizations associated with Ph-like ALL. Though some treatments have been shown to be successful for Ph-like patients, it is crucial for more clinical trials to study the effectiveness of specific TKIs on different genomic alterations with chemotherapy to improve clinical outcomes.

## Figures and Tables

**Figure 1 diagnostics-15-00385-f001:**
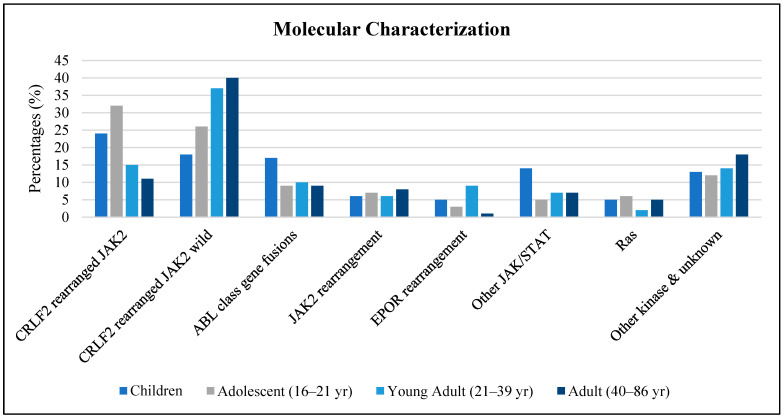
Data on genetic subtypes (adapted from [1]).

**Figure 2 diagnostics-15-00385-f002:**
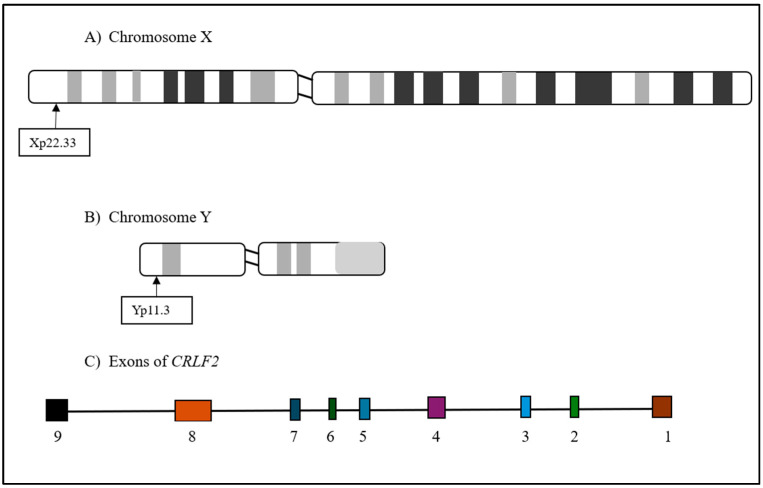
(**A**) Locus of *CRLF2* gene on Xp22.33; (**B**) locus of *CRFL2* gene on Yp11.3; (**C**) diagram of *CRLF2* gene on Xp22.33 and Yp11.3 (adapted from [7]).

**Figure 3 diagnostics-15-00385-f003:**
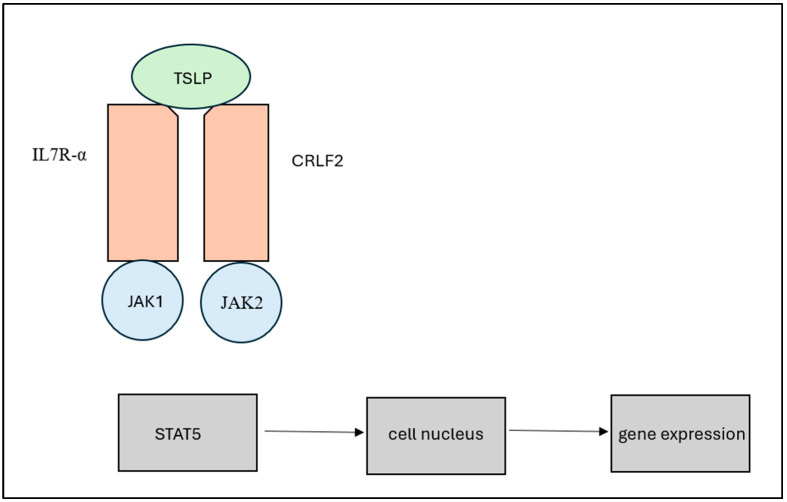
Mechanism of *CRLF2* (adapted from [12]).

**Figure 5 diagnostics-15-00385-f005:**
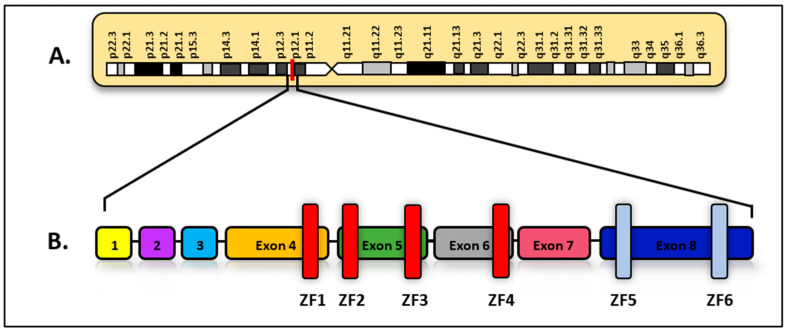
(**A**) Locus of *IKZF1* gene on 7p12.2; (**B**) diagram of the exon structure of *IKZF1* and its respective encoded zinc finger domains (ZF1–6) (adapted from [21]).

**Figure 6 diagnostics-15-00385-f006:**
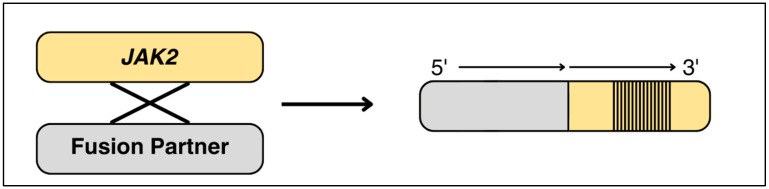
Fusion partners translocate to the 5′ end of the *JAK2* gene, causing continuous activation of the tyrosine kinase domain at the 3′ end of the *JAK2* gene (striped area) (adapted from [2]).

**Figure 7 diagnostics-15-00385-f007:**
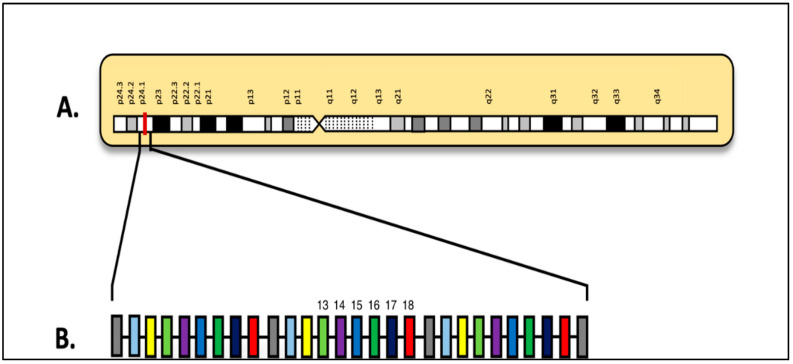
(**A**) Locus of *JAK2* gene on 9p24.1; (**B**) diagram of *JAK2* gene on 9p24.1 (adapted from [24,27]).

**Table 1 diagnostics-15-00385-t001:** Prevalence of Ph-like ALL in B-ALL (adapted from [1]).

Age Group	Children (0–16 Years)	Adolescents (16–20 Years)	Younger Adults (21–39 Years)	Older Adults (40+ Years)
Prevalence	12%	21%	27%	20–24%

**Table 2 diagnostics-15-00385-t002:** *ABL* class genes and their fusion partners. Data were adapted from [2].

*ABL* Class Gene (Locus)	5′ Fusion Partner
*PDGFRA* (4q12)	*FIP1L1*
*PDGFRB* (5q32)	*ATF7IP*, *EBF1*, *ETV6*, *SNX29*, *SSBP2*, *TNIP1*, *ZEB2*, *ZMYND8*
*CSF1R* (5q32)	*MEF2D*, *SSBP2*, *TBL1XR1*
*ABL2* (1q25.2)	*PAG1*, *RCSD1*, *ZC3HAV1*
*ABL1* (9q34)	*CENPC*, *ETV6*, *FOXP1*, *LSM14A*, *NUP153*, *NUP214*, *RANBP2*, *RCSD1*, *SFPQ*, *SNX1*, *SNX2*, *SPTNA1*, *ZMIZ1*

**Table 3 diagnostics-15-00385-t003:** Fusion partners of *JAK2*, *EPOR*, and JAK/STAT.

Alteration Classification	Genes Involved	Frequency	Tyrosine Kinase Inhibitor
*JAK2*	*ATF7IP*, *BCR*, *EBF1*, *ETV6*, *GOLGA4*, *GOLGA5*, *HMBOX1*, *NPHP3*, *OFD1*, *PAX5*, *PCM1*, *PPFIBP1*, *RFX3*, *RNPC3*, *SMU1*, *SNX29*, *SPAG9*, *SSBP2*, *STRBP*, *STRN3*, *TERF2*, *TPR*, *USP25*, *ZBTB20*, *ZBTB46*, *ZEB2*, *ZNF274*, *ZNF340*	~7%	Ruxolitinib
Other JAK/STAT	*IL7R*, *IL2RB*, *JAK1*, *JAK3*, *SH2B3*, *FLT3*, *TYK2*	13%	Ruxolitinib (exceptions: TYK2 inhibtor for *TYK2* mutations, FLT3 inhibitor for *FLT3* mutations)
*EPOR*	*IGH*, *IGK*, *LAIR1*, *THADA*	~4%	Ruxolitinib

## Data Availability

Not applicable.

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
