# Peer review of "Integrative Insights into Philadelphia-like B-Cell Acute Lymphoblastic Leukemia: A Genetic and Molecular Landscape"

_diagnostics, 2025, doi:10.3390/diagnostics15030385_

Round 1

Reviewer 1 Report

Comments and Suggestions for Authors

The manuscript is comprehensive and well-written. I have some minor suggestions: 

1. In the first paragraph of section 4, the authors mentioned IKZF1 but did not illustrate more on this. Please add more information before the illustration on ABL genes. 

2. In the third paragraph of section 5, the authors mentioned that appearance of a mutation in multiple sub clones meaning it is the second driver genes for Ph-like ALL. Why the common mutation is not the first driver gene? 

3. Could the author describe the difference between Ph-like and BCR-ABL like ALL?

Author Response

Comment 1 : In the first paragraph of section 4, the authors mentioned IKZF1 but did not illustrate more on this. Please add more information before the illustration on ABL genes.

Response 1: DONE

Thank you for pointing this out. We agree with this comment. Therefore, we have added the percentage of Ph-like cases IKZF1 alterations can be found and expanded more on the function of IKZF1 and its loss which can be found in section 4, paragraph 1, sentences 3-11, highlighted in yellow. We have also created a diagram of IKZF1 and its exons on figure 5, also highlighted in yellow.

Comment 2 In the third paragraph of section 5, the authors mentioned that appearance of a mutation in multiple sub clones meaning it is the second driver genes for Ph-like ALL. Why the common mutation is not the first driver gene?
Response 2: DONE

Thank you for pointing out this area for clarification. We have added the sentences, “While these mutations are common and may help drive the cancer, they typically emerge later as the disease progresses and are not the initial cause of Ph-like ALL. As such, they are considered secondary driver events—which are more often associated with multi-subclonality—rather than primary ones,” to better explain this. This can be found in the second sentence of paragraph 3 of section 5, highlighted in yellow.

Comment 3 : Could the author describe the difference between Ph-like and BCR-ABL like ALL?

Response 3: DONE

We agree. Therefore, we have added the sentence “Although Ph-like ALL is known as BCR::ABL1-like ALL, Ph-like refers to a ALL with similar molecular characterization to Philadelphia positive (Ph+) ALL whereas BCR::ABL1-like ALL signifies that there is no BCR::ABL1 fusion. Most cases are concordant [2]. Herein this review, the term Ph-like will be used,” to define the terms in further detail which can be found in the second sentence of paragraph 1 of section 1, highlighted in yellow. In the rest of the review, we will only use the term “Ph-like” in the manuscript to maintain constant terminology.

We have also added the percentage of Ph-like cases IKZF1 alterations can be found and expanded more on the function of IKZF1 and its loss which can be found in section 4, paragraph 1, sentences 3-10, highlighted in yellow. We have also created a diagram of IKZF1 and its exons on figure 5, also highlighted in yellow.  

Reviewer 2 Report

Comments and Suggestions for Authors

This review aims to provide an in-depth discussion of the genetic and molecular characteristics of Philadelphia-like B-cell acute lymphoblastic leukemia (Ph-like B-ALL). The manuscript addresses a relevant and timely topic. I have some concerns for the authors as below.

1.      In the abstract part, I would like to recommend the authors rewrite "Herein we are trying to review the current data on Ph-like ALL.” to "This review summarizes current findings on Ph-like ALL, focusing on its molecular landscape and clinical implications.”

2.      The manuscript uses both "Ph-like ALL" and "BCR::ABL1-like ALL". A consistent terminology should be maintained throughout.

3.      Some phrases need to be polished, such as “The CRLF2 gene is located on both the X and Y chromosomes and its transcription generates 9 exons”.

4.      The format of the current manuscript needs to be carefully refined, such as the text sizes and indentations.

5.      The references need to be carefully double-checked, such as references 14 and 15.

6.      References should follow a uniform format, preferably adhering to the journal's style guidelines.

Comments on the Quality of English Language

Some phrases can be improved to avoid misleading and confusion. 

Author Response

Comment 1 : In the abstract part, I would like to recommend the authors rewrite "Herein we are trying to review the current data on Ph-like ALL.” to "This review summarizes current findings on Ph-like ALL, focusing on its molecular landscape and clinical implications.”

Response 1: DONE

Thank you for pointing this out. We agree with this comment. Therefore, we have changed "Herein we are trying to review the current data on Ph-like ALL.” to “This review summarizes current findings on Ph-like ALL, focusing on its molecular landscape and clinical implications.” which can be found in the last sentence of the abstract on page 1, highlighted in yellow.

Comment 2 : The manuscript uses both "Ph-like ALL" and "BCR::ABL1-like ALL". A consistent terminology should be maintained throughout.

Response 2: DONE

We agree. Therefore, we have changed two terms previously listed as “BCR::ABL1-like ALL” to “Ph-like ALL” to maintain consistency. This change can be found in section 3, paragraph 2, line 1 and section 6, paragraph 3, line 2 both of which are highlighted in yellow.

Comment 3 : Some phrases need to be polished, such as “The CRLF2 gene is located on both the X and Y chromosomes and its transcription generates 9 exons”.

Response 3: DONE

We agree. Therefore, we have edited the phrase to “The CRLF2 gene, located on chromosomes X and Y, generates 9 exons after transcription” which can be found in section 3, paragraph 1, sentence 2 highlighted in yellow.

Comment 4 : The format of the current manuscript needs to be carefully refined, such as the text sizes and indentations.

Response 4: DONE.

We agree. Therefore, we have made the text sizes uniform across the manuscript. We have changed the table captions to size 10 font matching the rest of the manuscript. We have also indented any paragraphs that did not have an indentation before.

Comment 5: The references need to be carefully double-checked, such as references 14 and 15
Response 5: DONE

We agree. Therefore, we have changed. We have followed the instructions of the journal. The referenced are mentioned correctly in the body of the manuscript and also in the LIST OF REFERENCES. 

Comment 6: References should follow a uniform format, preferably adhering to the journal's style guidelines.
Response 6: DONE

We agree. Therefore, we have edited all references to be listed by Author, Manuscript title. Journal Year, Volume, page number. We have used the guidelines of the Journal. Some of the reference numbers were changed from the previous edition of our manuscript because we have added two new references.

We have added also the percentage of Ph-like cases IKZF1 alterations can be found and expanded more on the function of IKZF1 and its loss which can be found in section 4, paragraph 1, sentences 3-10, highlighted in yellow. We have also created a diagram of IKZF1 and its exons on figure 5, also highlighted in yellow.